# Few-femtosecond passage of conical intersections in the benzene cation

M.C.E. Galbraith[1], S. Scheit[2], N.V. Golubev[2], G. Reitsma[1], N. Zhavoronkov[1], V. Despré[2], F. Lépine[3], A.I. Kuleff[2], M.J.J. Vrakking[1], O. Kornilov[1], H. Köppel[2] & J. Mikosch[1]

Observing the crucial first few femtoseconds of photochemical reactions requires tools typically not available in the femtochemistry toolkit. Such dynamics are now within reach with the instruments provided by attosecond science. Here, we apply experimental and theoretical methods to assess the ultrafast nonadiabatic vibronic processes in a prototypical complex system—the excited benzene cation. We use few-femtosecond duration extreme ultraviolet and visible/near-infrared laser pulses to prepare and probe excited cationic states and observe two relaxation timescales of $11 \pm 3$ fs and $110 \pm 20$ fs. These are interpreted in terms of population transfer via two sequential conical intersections. The experimental results are quantitatively compared with state-of-the-art multi-configuration time-dependent Hartree calculations showing convincing agreement in the timescales. By characterising one of the fastest internal conversion processes studied to date, we enter an extreme regime of ultrafast molecular dynamics, paving the way to tracking and controlling purely electronic dynamics in complex molecules.

[1] Max-Born-Institut, Max-Born-Strasse 2A, Berlin 12489, Germany. [2] Physikalisch-Chemisches Institut, Universität Heidelberg, Im Neuenheimer Feld 229, Heidelberg 69120, Germany. [3] Institut Lumière Matière, Université Lyon 1, CNRS, UMR 5306, 10 Rue Ada Byron, Villeurbanne Cedex 69622, France. Correspondence and requests for materials should be addressed to J.M. (email: jochen.mikosch@mbi-berlin.de)

The advent of attosecond technology[1] has opened new frontiers in contemporary physics and chemistry aimed at observing the motion of electrons in atoms, molecules and solids on their natural timescale[2]. In molecular physics, a lot of attention is currently centered around hole migration in excited molecular cations - an ultrafast propagation of charge through a molecular structure induced by the attosecond creation of a coherent superposition of suitable electronic states[3–8]. These dynamics are often considered as purely electronic, however in a real molecular system the electronic processes are not independent from accompanying nuclear dynamics. Recent theoretical studies have focused on how the motion of a nuclear wavepacket[9, 10] and nonadiabatic effects[11] lead to decoherence of hole wavepackets in the benzene molecule and benzene derivatives. Motivated by this work, the present study investigates ultrafast coupled electronic-nuclear dynamics in the excited benzene cation with unprecedented time resolution both experimentally and theoretically.

Near conical intersections the natural timescales for electronic and nuclear motion become comparable, leading to a breakdown of the Born-Oppenheimer approximation. In this regime electronic and nuclear motion are strongly coupled[12]. The experimental realisation of extremely short extreme ultraviolet (XUV) pulses by means of high-order harmonic generation (HHG) using few-cycle visible/near-infrared (VIS/NIR) pulses sparked recent experimental and theoretical investigations of such dynamics in cationic molecular systems, for instance in $CO_2^+$[13] and small hydrocarbon ions[14–16]. An accurate description of coupled electronic and nuclear dynamics remains however a formidable theoretical challenge, as it requires all nonadiabatic couplings between potential energy surfaces to be taken into account. The benzene cation is a prototypical complex system in which non-adiabatic dynamics are expected on the few-femtosecond timescale, as a number of conical intersections are located close to the Franck-Condon region. Its dynamics thus became a focal point for modelling, in particular for the multi-configuration time-dependent Hartree (MCTDH) method[17]. In a series of publications, one of the co-authors and his coworkers have theoretically investigated the vibronic coupling in the benzene cation[18–20] and its substitutes[21–25]. This benchmark multistate, multimode treatment represented the first quantum dynamical investigation of a molecule exceeding three coupled potential energy surfaces.

Electronic structure calculations within the linear vibronic coupling model were used to evaluate the potential energy surfaces and relevant curve crossings for the lowest five cationic states $\tilde{X}$-$\tilde{E}$ in the benzene cation, as schematically depicted in Fig. 1[19]. The $\tilde{X}^2E_{1g}$, $\tilde{B}^2E_{2g}$, and $\tilde{D}^2E_{1u}$ states are doubly degenerate, leading to eight component states in total. All of the electronic states investigated are interconnected by low-energy conical intersections. MCTDH simulations established that population in the $\tilde{E}^2B_{2u}$ state is transferred to the $\tilde{D}$ state and subsequently to the $\tilde{B}$ state through two conical intersections that can be reached without significant energy barriers. Note, the $\tilde{D}$- and $\tilde{C}^2A_{2u}$-state potentials intersect at high energies and far away from the equilibrium geometry of the cation (cf. Fig. 1) and thus $\tilde{D}$-$\tilde{C}$ internal conversion is not expected to be a relevant decay channel. The lower lying $\tilde{C} \rightarrow \tilde{B} \rightarrow \tilde{X}$ states show a similar internal conversion pathway upon initial population of the $\tilde{C}$ state[18, 20].

These in-depth theoretical studies are contrasted by a lack of validating time-resolved experiments. Until recently such studies were experimentally out of reach. To prepare the lowest cationic states from the neutral molecule, photon energies in the XUV frequency range are required. In addition, for experiments to be meaningful, a time resolution of a few femtoseconds is necessary, due to the ultrashort excited-state lifetimes involved.

Here we present a time-resolved experimental study combined with an extended theoretical treatment, which validates the application of the MCTDH theory for the complex dynamics of ultrafast relaxation in the benzene cation. We prepare a superposition of the lowest five electronic states with a spectrally filtered few-femtosecond XUV pulse. The internal conversion dynamics are probed by further exciting the prepared cations with a delayed few-cycle VIS/NIR probe pulse and by measuring specific fragment yields as a function of the XUV-VIS/NIR delay. MCTDH calculations of the internal conversion are performed that are tailored to the experimental conditions, with wavepackets initiated in the relevant $\tilde{E}$ and $\tilde{D}$ cationic states, and are quantitatively compared to the experimental results.

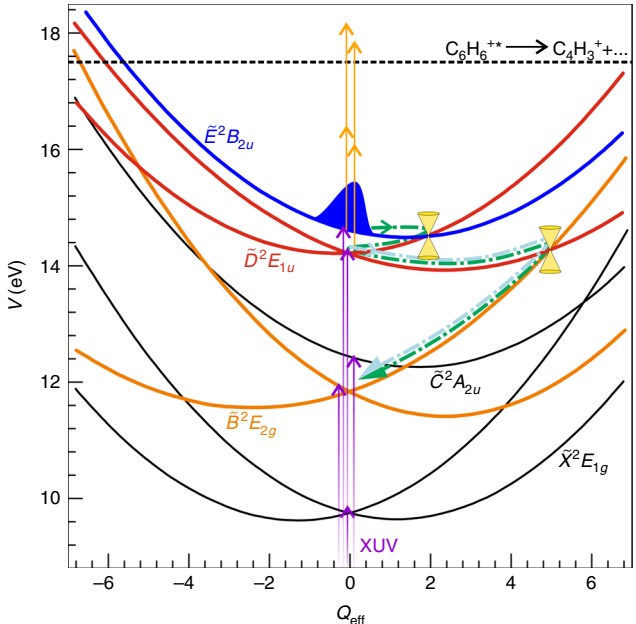

**Fig. 1** Schematic of the studied dynamics. Schematic overview of the lowest eight electronic component states of the benzene cation, depicted as potential energy $V$ in eV as a function of a dimensionless effective nuclear coordinate $Q_{eff}$[19] (see also Supplementary Note 1). Ionisation by the XUV pulse can simultaneously populate all of the cationic states that are shown (violet arrows). Of these cationic states, only the $\tilde{D}$-state and the $\tilde{E}$-state can be excited beyond the threshold for $C_4H_3^+$ formation[26] by a two-photon process (orange arrows). The dashed green and light blue curves are a cartoon drawing of the time-evolution of a cation originally transferred to the $\tilde{E}^2B_{2u}$ and to the $\tilde{D}^2E_{1u}$ state, respectively. Internal conversion processes via the conical intersections indicated in the figure lead to population of the $\tilde{B}^2E_{2g}$ state. The potential energy surfaces are reproduced from ref. [19]

## Results

**Preparation of benzene cations in excited states.** The XUV pulses in the experiment were produced by HHG in atomic xenon and filtered with a thin indium metal foil (see Methods section). They consist mainly of the 9th harmonic. Its spectrum is centered around a photon energy of $\hbar\omega = 16.0$ eV as displayed in Fig. 2a. Also shown in this figure are partial photoionisation cross sections of benzene, as previously determined by angle-resolved photoelectron spectroscopy[27]. The lowest five cationic electronic states are prepared with significant population in our experiment. For preparation of the lower-lying $\tilde{X}$, $\tilde{B}$, and $\tilde{C}$ electronic states, the photoelectron carries away a significant part of the photon energy[28]. The highest-lying states, $\tilde{E}$ and $\tilde{D}$, are prepared with moderate vibrational excitation[28]. Note that compared to the

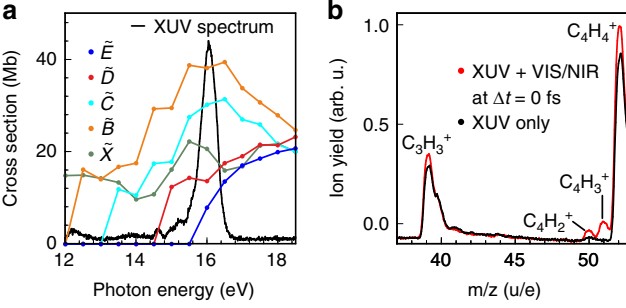

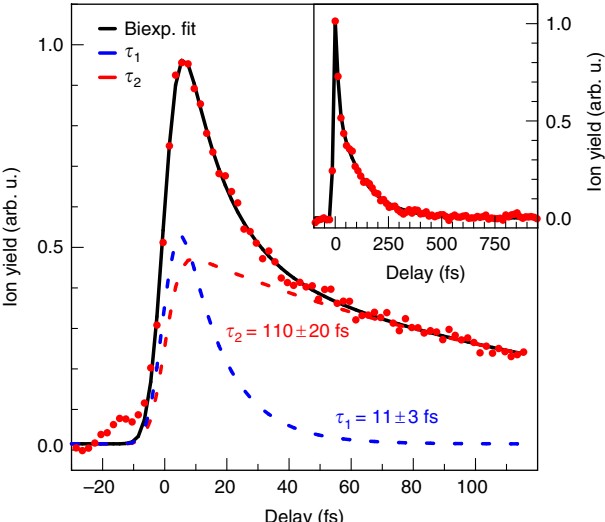

**Fig. 2** XUV and mass spectrum. **a** Spectrally filtered XUV spectrum (solid black line), as used in the experiment, shown alongside partial photoelectron cross sections for the first five cationic states of benzene (taken from ref. [27]). **b** The mass spectrum acquired for time-overlapped XUV pump and VIS/NIR probe pulses ($\Delta t = 0$ fs, red line) exhibits the appearance of $C_4H_2^+$ and $C_4H_3^+$, which are not observed in an XUV-only (black line) measurement

**Fig. 3** Time-resolved experimental data. Experimentally measured $C_4H_3^+$ fragment yield as a function of the XUV-VIS/NIR delay (red dots). The bold black line is a biexponential fit to the data, the dashed lines represent the contributions from the two timescales. The inset displays a long range pump-probe scan of $C_4H_3^+$

ionisation cross sections, the cross sections for producing superexcited neutral states are negligible[29].

Ionisation of benzene with the present XUV spectrum leads to fragmentation as the photon energy exceeds the appearance energy for unimolecular dissociation of $\hbar\omega > 13.8$ eV[26]. Five fragments are observed in the mass spectrum (see Supplementary Note 2), of which the most relevant section is displayed in Fig. 2b (black line), in agreement with tabulated ion appearance energies[26]. The fragmentation processes in benzene cations have previously been characterised as statistically driven, occurring after relaxation to the electronic ground state (see refs [26, 30–32] and references therein). Therefore the observed fragmentation patterns are determined by the total internal energy of the cations upon photoionisation.

**Time-resolved probing of dynamics in benzene**. Probing of the XUV-prepared benzene cations with a VIS/NIR pulse at time-overlap leads to the appearance of two additional fragments, $C_4H_2^+$ and $C_4H_3^+$ (see mass spectrum in Fig. 2b (red line)). These two fragments also originate from unimolecular dissociation of benzene cations and have appearance energies of $\hbar\omega = 17.5$ eV[26], i.e. above the upper edge of the XUV spectrum in Fig. 2a. Note that double ionisation does not play a role in these experiments as the second ionisation potential is 24.9 eV[33]. We thus conclude that the VIS/NIR pulse induces cation-cation transitions from electronic states initially prepared by the XUV. This allows us to investigate the relaxation dynamics of cations produced by the XUV pulse by recording two-colour mass spectra as a function of the pump-probe time delay $\Delta t$. The yield of $C_4H_3^+$ fragments as a function of time delay is presented in Fig. 3 A positive delay corresponds to the XUV pulse arriving first at the sample. An increase of the fragment yield around time-overlap is observed, followed by an ultrafast decay. At long delays ($\geq 500$ fs) the yield is found to decrease to the level observed before $\Delta t = 0$ (cf. inset in Fig. 3). The small contribution observed at negative time delay in Fig. 3 probably originates from imperfect dispersion compensation of the VIS/NIR pulse, resulting in a small post-pulse.

The transient in Fig. 3 cannot be described by a single exponential decay. Instead, it is found that a fit with a biexponential decay function, convoluted with a Gaussian function which represents the finite duration of the laser pulses, resembles the data excellently (black line). The two contributions to the fit have similar amplitudes and are shown as dashed lines

in the figure. Time constants of $\tau_1 = 11 \pm 3$ fs and $\tau_2 = 110 \pm 20$ fs for the two decay processes are derived from the fit. The full width at half maximum of the Gaussian function is also found from the fit and amounts to $\tau_{IRF} = 10 \pm 1$ fs. As discussed below this instrument response function reflects a process involving the absorption of one XUV and two VIS/NIR photons dominant in our experimental signals. The fragment $C_4H_2^+$ exhibits similar biexponential dynamics, but with an inferior signal-to-noise level. The decay times are consistent with those for the $C_4H_3^+$ fragment. Furthermore, we find the decay times not to vary with the VIS/NIR probe intensity.

It is instructive to determine the number of photons that are absorbed in the probe step. For this purpose two-colour mass spectra were recorded as a function of the probe intensity. As the pump-probe signals of $C_4H_2^+$ and $C_4H_3^+$ consist of contributions featuring different decay rates and comparable amplitudes, measurements were carried out both at time-overlap ($\Delta t = 0$ fs) as well as at $\Delta t = 80$ fs. Linear fits of the probe power-dependent fragment yields in a double logarithmic plot result in slopes of $m_{\Delta t = 0\,fs} = 2.0 \pm 0.3$ and $m_{\Delta t = 80\,fs} = 2.1 \pm 0.1$ for the fragment $C_4H_3^+$ (see also Supplementary Note 3). These two values show that both contributions of the biexponential decay originate from a two-photon probing mechanism.

**Discussion**

A benzene cation needs to possess a total energy of at least 17.5 eV above the neutral ground state in order for fragmentation producing $C_4H_3^+$ and $C_4H_2^+$ to be possible (see Fig. 1)[26]. As indicated by the experiment, the probe laser contributes the energy of two VIS/NIR photons ($\approx 3.2$ eV) to this cationic energy. Thus, it follows that the initial energy of the cation produced by the XUV ionisation must be at least 14.3 eV. Among the five aforementioned electronic states, only the $\tilde{E}$ and $\tilde{D}$ states satisfy this requirement. We thus attribute the observed time-dependent $C_4H_2^+/C_4H_3^+$ fragment signals to the evolution of populations in the $\tilde{E}$ and $\tilde{D}$ electronic states following XUV pumping.

The efficiency of fragment formation from the $\tilde{E}$-state population is expected to be different from that of the $\tilde{D}$-state

population, as it depends on different VIS/NIR-induced cation-cation transitions (see Supplementary Note 4). Note that in contrast to time-resolved photoelectron spectroscopy[34], the VIS/NIR probe step has to be resonant in the present case, which is facilitated by the broad frequency spectrum of the probe pulse. When the XUV-excited cations relax by internal conversion, energy is transferred from electronic to vibrational degrees of freedom. In particular internal conversion from the $\tilde{D}$ to the low-lying $\tilde{B}$ state is expected to sharply reduce the probe-induced signal as a lot of energy is transferred from electronic to nuclear degrees of freedom (Fig. 1), leading to a delocalization of the vibrational wavepacket. Following the Condon principle, cation-cation probe transitions would then lead to highly vibrationally excited levels. The limited availability of suitable electronic states in the relevant energy range (i.e. with vibronic ground states at $\approx 11.6\,\text{eV} + 2 \times 1.6\,\text{eV} = 14.8\,\text{eV}$) renders such transitions unlikely (cf. Fig. 1 main text and Supplementary Fig. 1a). Calculated transition probabilities from the $\tilde{B}$ to the $\tilde{D}$, $\tilde{E}$, $\tilde{F}$, and $\tilde{G}$ states are found to be all negligible (see Supplementary Note 5).

Note that in our experiment the lower-lying $\tilde{D}$ state is both directly populated by the XUV and indirectly through decay of the XUV-populated $\tilde{E}$ state. The rate equations for relaxation are solved by a double-exponential decay with the same time constants that would result from a pure preparation of the higher-lying $\tilde{E}$ state. We thus can attribute the two exponential time constants observed in the experiment as being associated with relaxation dynamics via the $\tilde{E} \rightarrow \tilde{D}$ and $\tilde{D} \rightarrow \tilde{B}$ conical intersections.

The assignments made above can be tested against MCTDH calculations capable of tackling the complex problem of benzene cation relaxation. Following the method described in ref. [18] (see Methods section for a brief description) MCTDH calculations were performed taking into account the $\tilde{E}\,^2B_{2u}$ state, the degenerate $\tilde{D}\,^2E_{1u}$ states and the degenerate $\tilde{B}\,^2E_{2g}$ states. Other low-lying states are expected to be irrelevant in the present conditions due to the two-photon probing mechanism established above. The previous results of ref. [18] were extended in two different ways. First, the initial electronic state was chosen both as the $\tilde{E}$ state and as the $\tilde{D}$ state of the cation. This is necessary because the pump laser creates both states in the experiment. Second and more important, in the previous work only diabatic electronic populations were considered because the use of diabatic electronic wavefunctions greatly facilitates quantum-dynamical computations; they were thus underlying the Hamiltonian of ref. [18]. Conceptually, however, adiabatic populations are more instructive because they underlie common thinking of how molecular motion proceeds and how molecular structure is defined. Adiabatic electronic wavefunctions are uniquely characterised through eigenvalues of the molecular potential energy operator (fixed-nuclei electronic hamiltonian) and are naturally obtained in the first place by quantum-chemical computations. We have hence also extracted adiabatic populations from our MCTDH calculations, which would have been impossible at the time of the earlier work reported in[18, 19] due to the great numerical cost. We will publish a more detailed description elsewhere and confine ourselves here to describing the comparison with the experimental data. A wavepacket initiated in the $\tilde{E}$ state is found to undergo internal conversion to the $\tilde{D}$ state within a fraction of the vibrational period ($\leq 20$ fs), whereas cations prepared in the $\tilde{D}$ state decay towards the $\tilde{B}$ state within a few hundreds of fs (see Supplementary Note 6). The calculated population dynamics thus further substantiate our interpretation that the ultrafast timescale of $\tau_1 = 11 \pm 3$ fs, observed in the experiment, originates from the decay of $\tilde{E}$ state population via the $\tilde{E} \rightarrow \tilde{D}$ conical intersection while the observed slower decay process with $\tau_2 = 110 \pm 20$ fs

originates from the decay of $\tilde{D}$ state population via the $\tilde{D} \rightarrow \tilde{B}$ conical intersection. In the calculation, a considerable fraction of population remains trapped in the $\tilde{D}$ state, in contrast to the known absence of fluorescence in the benzene cation[35]. This trapped population is an artefact both of the employed diabatic state basis and of unaccounted vibrational modes. The trapped population gets smaller in the adiabatic representation and has been removed before further analysis.

It should be noted that the link between the electronic state populations and the experimental observables is indirect. Nonetheless we argue that the transient behaviour of the experimental signals is mainly influenced by population dynamics and thus the experimental and theoretical timescales can be compared. The XUV pulse prepares both the $\tilde{E}$ and $\tilde{D}$ states of the cation. Therefore we construct a superposition of the two aforementioned wavepacket calculations and compare it with the experiment in the following way: the fractional initial populations of the $\tilde{D}$ and $\tilde{E}$ cation states upon XUV ionisation are obtained as $p_0(\tilde{E}) = 36\%$ and $p_0(\tilde{D}) = 64\%$ from Fig. 2b by convoluting the respective partial photoionisation cross sections of these two states with the measured XUV pump spectrum. The total time-dependent $\tilde{E}$ and $\tilde{D}$ state populations are hence calculated by adding the respective populations from the $\tilde{E}$- and $\tilde{D}$-state-initiated wavepacket calculations, using weights of 36% and 64%, respectively. Finally, the total $\tilde{E}$ and $\tilde{D}$ state populations are added, weighted by a relative probing efficiency of $\tilde{E}$ and $\tilde{D}$, $\text{Eff}_{\tilde{E}}/\text{Eff}_{\tilde{D}}$, to obtain a theoretical 'fragment yield' curve. $\text{Eff}_{\tilde{E}}/\text{Eff}_{\tilde{D}}$, which incorporates the overall relative efficiency to form $C_4H_3^+$ fragments in the probe step, is the only adjustable parameter. In our analysis, we ignore any potential influence of vibrational dynamics within the $\tilde{E}$ and $\tilde{D}$ states on the probing efficiencies. We justify this approximation by the extensive averaging over the coordinate-dependence of Franck-Condon factors due to the many internal coordinates involved. The fragment formation efficiency is determined by the two-photon cation-cation transition probabilities, weighted with the spectral intensity of the probe pulse, and by the final state-dependent relative probability of $C_4H_3^+$ formation. A detailed analysis of the probe step, including computed cation-cation transition probabilities, is presented in Supplementary Note 5 (see Supplementary Fig. 1). We find that while more final states are being accessed with significant probability from the $\tilde{D}$ state, the transitions originating from the $\tilde{E}$ state lead to higher-lying final states, from which the probability for dissociation towards $C_4H_3^+$ is higher. The amplitudes of the two contributions in the experimentally measured pump-probe trace indicate a value of $\text{Eff}_{\tilde{E}}/\text{Eff}_{\tilde{D}} = 2$. This ratio is consistent with our computation of the relevant cation-cation transition moment amplitudes (see Supplementary Note 5).

The theoretical 'fragment yield' curve resulting from this summation is depicted in Fig. 4. The solid light blue line represents the result from the adiabatic calculation, while the dashed dark blue line represents the result from the diabatic calculation. Adiabatic and diabatic results are similar and well described by a biexponential fit, which yields time constants of $\tau_1 = 8/12$ fs and $\tau_2 = 170/190$ fs (adiabatic/diabatic). These results are weakly dependent on the assumed relative probing efficiency of the $\tilde{D}$ and $\tilde{E}$ states. Variation of $\text{Eff}_{\tilde{E}}/\text{Eff}_{\tilde{D}}$ within limits that reasonably resemble the experimental behaviour leads to decay times in the range of $\tau_1 = 6\text{-}11/11\text{-}15$ fs and $\tau_2 = 160\text{-}180/180\text{-}200$ fs (adiabatic/diabatic). An oscillation of the yield derived from the diabatic calculation with a period of about 35 fs is visible, which is absent in the adiabatic result, as well as a pronounced maximum at about 50 fs. These features likely emerge from variations of the adiabatic-to-diabatic mixing angle and are considered to be an artefact of the diabatic theory, observed also in previous two-state

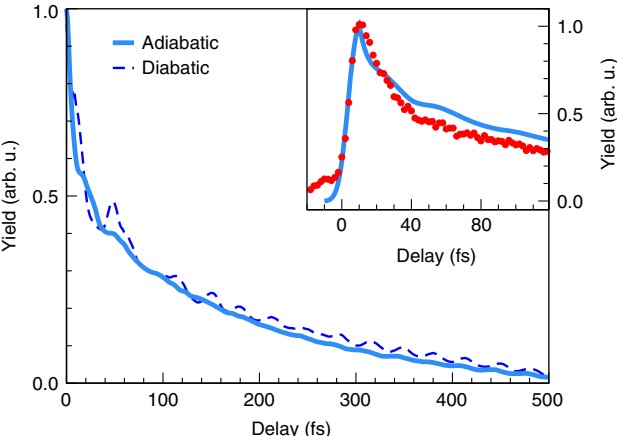

**Fig. 4** Comparison of theory and experiment. Time-dependent 'yields' (blue lines), obtained by summing the populations of the $\tilde{E}$ and $\tilde{D}$ states from MCTDH calculations (see main text). The curves are derived from separate wavepacket calculations for an initial pure excitation in the $\tilde{E}$ and in the $\tilde{D}$ state, respectively. The solid (dashed) line represent the result obtained for adiabatic (diabatic) electronic populations. The inset displays the yield for the adiabatic state calculation, convoluted with the Gaussian instrument response function, for direct comparison to the experiment (red points, cf. Fig. 3)

calculations[36]. The theoretical 'yield' curve for the adiabatic calculation is further convoluted with the Gaussian instrument response function of $\tau_{IRF} = 10$ fs and plotted on top of the experimental result in the inset of Fig. 4. The agreement between experiment and theory is striking, strongly supporting the quantitative validity of the MCTDH calculations. We note that a larger variation at longer time scales is not unexpected since only a selection of vibrational states is included in the simulation. This observation might also hint at a potential coordinate-dependence of the transition operator associated with the experimental probe step. We emphasize that this is the first experimental verification of these large-scale computational results, originally published as benchmark calculations and as a purely theoretical prediction.

The comparative study presented here demonstrates that complex nonadiabatic relaxation processes inevitably accompany - already at very short times - the purely electronic dynamics sought after in charge-migration experiments. These relaxation processes may span several electronic states, coupled by many vibrational degrees of freedom. In the present case of the benzene cation, the $\tilde{E} \rightarrow \tilde{D}$ internal conversion limits the purely electronic part of the dynamics to a timescale of about 10 fs, after which time nuclear dynamics and the associated dephasing of the electronic wavepackets cannot be neglected (see also refs 13, 37). To separate these two effects, experimental and theoretical access to the time-evolution of the involved electronic states is crucial. We demonstrated that monochromatized short XUV laser pulses combined with few-cycle VIS/NIR laser pulses allow for a state-by-state analysis of the population evolution. Moreover, we established that MCTDH calculations are suitable to predict these population evolutions. Our experimental and theoretical approach combined with the ongoing development of frequency-tunable few-femtosecond XUV sources[38, 39] should serve as a benchmark for future experiments towards the observation of attosecond dynamics.

In conclusion, we have experimentally determined decay rates of cationic states of benzene that are produced by XUV ionisation of the neutral molecule. Two timescales of $\tau_1 = 11 \pm 3$ fs and $\tau_2 = 110 \pm 20$ fs are observed and interpreted as $\tilde{E} \rightarrow \tilde{D}$ and

$\tilde{D} \rightarrow \tilde{B}$ internal conversion timescales, respectively. The $\tilde{E}$ state relaxation timescale marks one of the fastest internal conversion processes observed to date in a time-resolved experiment.

## Methods

**Experimental methods.** The experiments were carried out with a two-colour XUV-VIS/NIR pump-probe setup, which has already largely been described elsewhere[40–42]. The experimental apparatus was upgraded for the present study by compressing the output pulses of a 1 kHz Ti:Sa amplifier by means of a hollow-core fibre setup and a set of chirped mirrors. Few-cycle pulses with a bandwidth of 640-920 nm (FWHM) and a pulse duration of $\tau = 6$ fs, determined with a SEA-F-SPIDER measurement[43], were obtained. The beam is split into two parts. The first part is used to generate high harmonic radiation in a gas cell containing xenon. The resulting broadband XUV spectrum is then filtered with a 200 nm thick indium foil. This limits the spectral bandwidth[44] and results in the spectrum depicted in Fig. 2a. The XUV photon energy was calibrated by measuring the absorption spectrum of molecular nitrogen (see Supplementary Note 7). The XUV beam was recombined with the second part of the VIS/NIR beam, which acts as a probe, by use of a cored mirror. Using glass wedges for dispersion control, the two arms of the setup were then fine-tuned: (i) the VIS/NIR arm was optimised using argon strong field ionisation yield, and (ii) the XUV arm was optimised using HHG efficiency and spectrum. This allowed us to reach a situation were essentially a single harmonic was transmitted through the Indium metal filter, which was not significantly spectrally cut. The two recombined, collinear beams are then focused by a toroidal mirror into the center of a velocity-map-imaging spectrometer, operated in time-of-flight mode[45]. The benzene sample was provided by a thin supply pipe, with its orifice in close proximity to the interaction region. The VIS/NIR probe beam intensity was set to approximately $10^{12}$ W cm$^{-2}$. Due to the multiphoton nature of the molecular response and potentially not fully compensated pulse propagation effects, we determined a Gaussian instrument response function from a fit of the delay-time dependent ion yield. This resulted in a value of $\tau_{IRF} = 10 \pm 1$ fs. Note that our data cannot be fitted convincingly by the response function and a single exponential decay.

**Theory methods.** The time-dependent electronic populations used in this work are obtained in a fully quantal way with the aid of the Multiconfigurational Time-Dependent Hartree (MCTDH) method. This is a highly efficient tool for solving the time-dependent Schrödinger equation for several (or many) degrees of freedom. The standard method of solution would be the numerically exact propagation of a wavepacket represented in a time-independent product basis set. In the MCTDH scheme the wave function is instead written as a linear combination of Hartree products as follows:

$$\Psi(q_1, \ldots, q_f, t) = \sum_{j_1}^{n_1} \ldots \sum_{j_f}^{n_f} A_{j_1 \ldots j_f}(t) \prod_{k=1}^{f} \phi_{j_k}^{(k)}(q_k, t). \quad (1)$$

Here $f$ is the number of degrees of freedom, $q_1,\ldots,q_f$ are a set of nuclear coordinates, $A_{j_1 \ldots j_f}$ denote the time-dependent expansion coefficients, and $\phi_{j_k}^{(k)}(q_k, t)$ are a set of time-dependent single-particle functions (SPFs) combined to the Hartree product $\Phi_J$. In turn, the SPFs are expressed in a time-independent basis set:

$$\phi_{j_k}^{(k)}(q_k, t) = \sum_{i_1=1}^{N_k} c_{i_k}^{(k,j_k)}(t) \chi_{i_k}^{(k)}(q_k), \quad (2)$$

where $\chi_{i_k}^{(k)}(q_k)$ are primitive basis functions of the $k$-th degree of freedom that depend on the particle coordinate $q_k$. The accuracy of MCTDH depends on the number of primitive functions $N_1,\ldots,N_f$ and the number of SPFs $n_1,\ldots,n_f$ used. Since both the coefficients and the basis functions in Equation (1) are time-dependent, they both are optimised using a variational principle. The equations of motion are then derived from the Dirac-Frenkel variational principle[46, 47]. This choice keeps the product representation of the wavepacket optimally short and reduces the length of the state vector by about six orders of magnitude in typical applications (MCTDH contraction effect).

The implementation proceeds along the lines of ref. 18 where a model Hamiltonian for the $\tilde{B}$-$\tilde{D}$-$\tilde{E}$ states of $C_6H_6^+$ was set up within a linear vibronic coupling scheme and the ionisation potentials, vibrational frequencies, and coupling constants were determined from ab initio calculations. Up to nine nuclear degrees of freedom have been considered in the calculation, including the strongly Condon-active C-C stretching mode $\nu_2$ ($a_{1g}$ symmetry) and the Jahn-Teller active modes $\nu_{16}$-$\nu_{18}$ ($e_{2g}$ symmetry). In view of the degeneracy of the $\tilde{B}$ and $\tilde{D}$ states five component electronic states (five potential energy surfaces) have been treated. All basis set details have been collected in Tab. IV.b of ref. 18, which shows that $10^{11}$ primitive basis functions are reduced to $10^6$ time-dependent basis functions by the MCTDH contraction. The resulting electronic populations of the interacting states are depicted in Fig. 8b of ref. 18 and have been extended in different ways.

The wave packet has been propagated for 500 fs (i.e. much longer that the 200 fs of ref. [18]) and also for the $\tilde{D}$ component states as initial state (while in ref. [18] only the $\tilde{E}$ state was considered as initial state). In addition to the diabatic populations also the adiabatic ones have been computed (see Supplementary Note 6 for further details): the adiabatic representation is the more natural one to be used for the interpretation of the nuclear dynamics (for reasons given above). Our simulations allow to obtain a suitably weighted mean of the detected $\tilde{D}$ and $\tilde{E}$ state populations (see above) and hence to implement the proper initial conditions in line with the present experiment.

**Data availability**. The data that support the findings of this study are available from the corresponding author upon request.

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

## Acknowledgements

S.S. would like to acknowledge the Brigitte-Schlieben Lange program of the Ministry of Science, Research and Art of Baden-Württemberg for financial support. G.R.

thanks the Netherlands Organization for Scientific Research (NWO) for financial support (Rubicon 68-50-1410). V.D. and A.I.K. acknowledge the financial support by DFG through QUTIF Priority Programme. N.V.G. thanks IMPRS-QD for financial support.

## Author contributions

M.C.E.G., G.R., N.Z., F.L., M.J.J.V., O.K., J.M. designed the experiment. M.C.E.G., G.R., O.K., J.M. performed the measurements. S.S., H.K. performed the MCTDH simulations. N.V.G., V.D., A.I.K. performed the calculations on the cation-cation transitions. M.C.E.G., S.S., A.I.K., O.K., H.K., J.M. wrote the manuscript. All authors discussed the results and contributed to the manuscript.

## Additional information

**Competing interests:** The authors declare no competing financial interests.

