## [Peer Review File · Nature Communications]

Reviewers' comments:

Reviewer #1 (Remarks to the Author):

Manuscript ID: NCOMMS-17-04666-T

Title: "Few-Femtosecond Passage of Conical Intersections in the Benzene Cation"

In their work M. C. E. Galbraith and coworkers present an interesting investigation of ultrafast molecular relaxation that follows ionization by XUV radiation. They use harmonic radiation to ionize benzene and probe the relaxation of the cationic excited states with a short VIS/IR pulse. They observe fast dynamics in one particular fragmentation channel leading to the production of C₄H₃⁺ ions. Comparison with theoretical calculations allows them to identify the electronic excited states and the conical intersections involved in the relaxation process. To follow the evolution of an excited electron wave-packet on few-femtosecond time scales for a relatively big molecule such as benzene, it is not an easy task. In the present case, the authors manage to identify two particular conical intersections which are responsible for two different time scales: ~ 11 fs and ~ 110 fs. For this reason, they claim that this is the first experimental verification of large-scale computational results. The paper is well written. The work is properly presented and the results are convincing. So this manuscript deserves for sure publication. Nevertheless, I am not convinced that this work contains enough novel and relevant material for the broad readership of Nature Communications. On one hand, extending pump-probe measurements based on high-order harmonic radiation to few femtosecond time scales is not completely new. The authors themselves cite papers where even faster dynamics have been observed (Ref. 5). On the other hand, the explanation of the molecular dynamics is based on theoretical models and results which were already published (Ref. 16). For these reasons I consider the paper too incremental and technical for Nature Communications. I therefore suggest the authors to submit to a more specific journal once they replied to my comments listed here below.

Comments:

1) I found the introduction a bit too focused on benzene if the target is fast molecular dynamics. There are other groups who studied ultrafast dynamics of the first cation of (smaller) organic molecules. Recent examples are the theoretical work from the group of T. J. Martínez (e.g. Joalland, B et al, J. Phys. Chem. Lett. 2014, 5, 1467–1471) or the experimental work from the group of U. Keller (A. Ludwig et al, J. Phys. Chem. Lett. 2016, 7, 1901–1906). I feel that the authors should include more examples in their introduction in order to put their work in a better context and give a more exhaustive picture of the existing literature.

2) The authors derive the experimental cross-correlation width by fitting the experimental data. They state at page 3 that they used a "biexponential decay function, convoluted with a Gaussian function, representing the experimental time resolution,...". Later they write that "The experimental cross-correlation width is also found from the fit and amounts to $\tau_{xc} = 10 \pm 1$ fs." How do they extract the value of pump-probe cross correlation from the Gaussian width? Or in other words, how do they estimate the impulsive response of the system? Since they show that the probe process is a two-photon process, what the authors get is not strictly a cross-correlation between pump and probe. Can they provide an independent measurement for that?

3) I understand that the Indium filter cuts the radiation at energies above 17 eV in figure 2.a), but why there is no harmonic ~ 3 eV below the 9th harmonic in the figure? From the methods one sees that the authors used a Ti:Sa laser which normally is centered around 800 nm. Which is the central wavelength in this case? I ask this because if I assume a harmonic energy of ~ 16 eV for the 9th order I get a central wavelength of ~ 697 nm which is very close to the left border of their IR spectrum (640–920 nm). Is this correct? Could it be that the indium filter cuts the high energy part of the harmonic thus shifting its center of mass down to 16 eV? In this case, is it the 9th or the 11th harmonic that we are looking at? This could also explain the apparently small harmonic bandwidth. With a generating IR pulse of 6 fs one should get an even shorter envelope of the

generated attosecond pulse train. I would expect something around 3 fs. So the harmonics in the spectrum should look quite broad. Can the authors comment on this?

4) How do the authors calibrate the pump-probe delay? Do they use a different fragmentation channel or the same?

5) From the manuscript one understands that Q_{eff} is a vibrational coordinate. Actually which one? Can the authors give a more intuitive picture of how the nuclei are moving?

Reviewer #2 (Remarks to the Author):

This article describes new experiments which exploit ultrafast laser pulses in the XUV region to characterize the sub-picosecond dynamics of benzene cation. The primary importance of this work is that it demonstrates that even for very fast (<20fs) processes, one cannot neglect the nuclear motion. This is especially relevant given numerous discussions about ultrafast charge migration induced by attosecond pulses (which is often modeled theoretically without consideration of nuclear motion). The experiments are certainly state of the art. The agreement with theory is quite impressive, although the authors do make it clear that the interpretation (equating diabatic populations with experimental signals) is not entirely clear. Probably the most annoying part of this article is the paragraph where the authors praise the previous theoretical benzene cation work (last half of the second paragraph of the paper). Since one of the paper's authors is behind that work, this seems immodest at best. The authors should figure out how to discuss that work more objectively. Otherwise, the article is exceptionally well-done and is sure to stimulate further experiments and theory.

Reviewer #3 (Remarks to the Author):

This paper presents an impressive, state-of-the-art pump-probe mass spectrometry study of excited state dynamics in the benzene cation. An approx. 0.5 eV bandwidth 16 eV pump pulse prepares a set of cation states, ranging from X to E (over 5 eV). If transform limited, this pulse would be ~ 4fs in duration. A ~6fs NIR pulse is used as the probe and the fragmentation spectrum varies with time delay. In particular, the C₄H₃⁺ channel shows pump-probe contrast, as seen in Fig.2(b). The cross correlation is about 10 fs. Through a series of arguments, the authors claim that they probe only the higher E and D electronic states of the benzene cation and that they do so via a two-photon probe: i.e. the C₄H₃⁺ signal arises from a (1+2') process. A probe laser power study is used to support this claim. A state-of-the-art MCTDH calculation is used to model the results and the theoretical electronic population dynamics are compared to the (1+2') pump-probe signal. The agreement looks good, despite that there is no obvious direct link between the calculated population dynamics and the time-resolved C₄H₃⁺ fragment ion yield. Strong claims are made that the fragment ion yield maps the electronic population dynamics. However, this is not proven and could be just hopeful thinking.

There are several issues which need to be clarified before a recommendation for publication can be determined.

1. It took me some time to understand – I hope correctly - that the two time constants of 11 fs and 110 fs are for a parallel decay mechanism, meaning that the E and D states are prepared simultaneously (in a 1:2 ratio, they claim) but decay separately, each with their own time constant. If this is correct, then Fig.1 is highly misleading, as is some of the discussion. Fig. 1 leads the reader to believe that the E state decays to the D state which subsequently decays to the B-state. The reader thinks that the authors are reporting the probing of a sequential decay mechanisms E->D->B.

2. Given that the authors fit to two time constants to a parallel decay, where does the initial E-state population go? I assume to the D state, in which case there should be two components to the D-state decay. One direct preparation via D-state XUV absorption and then subsequent decay and a second path via E-state XUV absorption and then ultrafast decay to the D-state which then undergoes a slower decay. In other words, are there not both parallel and sequential components to the observed D-state decay?

3. In Fig.3, the pump-probe C4H3+ signal is presented and a biexponential fit is applied. Why is there signal before $t=0$? This is clearly seen in the first 20 fs on the 'negative' time delay side (red bump). Unless the pulse structures are not simple Gaussians as claimed, this means that, somehow, the NIR pulse is absorbed by the parent neutral molecule and, upon absorption of an XUV photon leads to a C4H3+ fragment. What is the mechanism by which this occurs? That this channel exists means that the analysis within the 10 fs cross correlation includes both pump-probe and probe-pump processes and that the fit to the 11 fs decay is not so straightforward.

4. The relationship between the calculated MCTDH electronic population dynamics and the time dependence of the (1+2') C4H3+ signal is not justified. I am worried that the apparent agreement may be accidental. What is the symmetry allowed two-photon transition from the E and D states? To which 'final' state? Is there only one? Are the two-photon oscillator strengths to the 'final' state the same for the E and D states? Does this not change the way in which the population dynamics should be compared to experiment?

5. How does the 'final' state prepared by (1+2') fragment to C4H3+? How do the Franck-Condon factors between the XUV prepared E and D-states and the 'final' state select the probing of the initial wavepacket? Prior studies by FF Crim et al used time-resolved photodissociation to probe wavepacket dynamics (see for example Journal of Chemical Physics 107, 661 (1997)) and interpreted their results in terms of the evolution of Franck-Condon overlaps. How can the authors avoid such effects? Does not vibrational dynamics in the E and D-states affect the C4H3+ yield? If so, then the C4H3+ yield cannot reflect only electronic population dynamics and the agreeable comparison with MCTDH populations is mostly lucky.

6. What are the estimated RRMK C6H6+ \rightarrow C4H3+ decay rates at the (1+2') level? Is it obvious that this fragment will appear rapidly enough in the extraction region to appear as a sharp mass peak? This would be yet another 'filtering' of the pump-probe signal. This needs better justification.

7. In Fig.4, inset, the MCTDH result (blue line) shows a bump at about 50 fs which is not seen at all in the experimental C4H3+ yield. Why is that? This should cause some concern that that experiment does not probe just the electronic population dynamics, as claimed.

Reviewers' comments:

Reviewer #1 (Remarks to the Author):

Manuscript ID: NCOMMS-17-04666-T

Title: "Few-Femtosecond Passage of Conical Intersections in the Benzene Cation"

In their work M. C. E. Galbraith and coworkers present an interesting investigation of ultrafast molecular relaxation that follows ionization by XUV radiation. They use harmonic radiation to ionize benzene and probe the relaxation of the cationic excited states with a short VIS/IR pulse. They observe fast dynamics in one particular fragmentation channel leading to the production of $C_4H_3^+$ ions. Comparison with theoretical calculations allows them to identify the electronic excited states and the conical intersections involved in the relaxation process. To follow the evolution of an excited electron wave-packet on few-femtosecond time scales for a relatively big molecule such as benzene, it is not an easy task. In the present case, the authors manage to identify two particular conical intersections which are responsible for two different time scales: ~ 11 fs and ~ 110 fs. For this reason, they claim that this is the first experimental verification of large-scale computational results. The paper is well written. The work is properly presented and the results are convincing. So this manuscript deserves for sure publication. Nevertheless, I am not convinced that this work contains enough novel and relevant material for the broad readership of Nature Communications. On one hand, extending pump-probe measurements based on high-order harmonic radiation to few femtosecond time scales is not completely new. The authors themselves cite papers where even faster dynamics have been observed (Ref. 5). On the other hand, the explanation of the molecular dynamics is based on theoretical models and results which were already published (Ref. 16). For these reasons I consider the paper too incremental and technical for Nature Communications. I therefore suggest the authors to submit to a more specific journal once they replied to my comments

listed here below.

We agree that the experimental demonstration of few-femtosecond dynamics using a high-harmonic source is not completely novel in itself, and that recently even faster dynamics have been reported in the paper by Calegari et al. (Ref. 5). However, the purpose of our manuscript is not reporting an experimental "first". Instead we show that with a combination of cutting-edge experimental and theoretical tools we can provide comprehensive insight into the few-femtosecond dynamics in a polyatomic molecule and that nuclear dynamics cannot be neglected on this timescale.

The theoretical treatment included in our original submission went beyond existing published results: A comparison with our experimental results required the diabatic state theory of Ref. 16 to be extended to multiple initial states prepared by the XUV pulse in the experiment. More importantly, to address the referee's concern, we have made a substantial revision to improve the manuscript: We have in the meantime completed computationally very expensive numerical computations to extract the adiabatic electronic populations from our MCTDH calculations. Adiabatic populations are preferred from a conceptual point of view but such reports are rare in the literature for cases where more than two coupled states are involved. Obtaining them is far from trivial, and their determination was considered impossible at the publication time of Refs. 14 and 16 due to computational cost. However, in the updated manuscript these adiabatic results are now included alongside the diabatic results that were already contained in the original submission. We will further elaborate on the details of the adiabatic computations in a subsequent manuscript to be submitted elsewhere. Note that the essential findings and conclusions of this manuscript are unaffected due to the overall good agreement of the two types of calculations. In detail, the agreement with the experiment is improved. Artefacts arising from the use of the diabatic state basis, some of which were pointed out by referee 3, are removed (see the updated Fig.4).

Comments:

1) I found the introduction a bit too focused on benzene if the target is fast molecular dynamics. There are other groups who studied ultrafast dynamics of the first cation of (smaller) organic molecules. Recent examples are the theoretical work from the group of T. J. Martínez (e.g. Joalland, B et al, J. Phys. Chem. Lett. 2014, 5, 1467–1471) or the experimental work from the group of U. Keller (A. Ludwig et al, J. Phys. Chem. Lett. 2016, 7, 1901–1906). I feel that the authors should include more examples in their introduction in order to put their work in a better context and give a more exhaustive picture of the existing literature.

We thank the referee for this remark. We have expanded the second paragraph of the manuscript to describe that the experimental realization of extremely short XUV pulses by means of HHG using few-cycle VIS/NIR pulses sparked recent experimental and theoretical investigations of non-adiabatic dynamics in cationic systems. We give examples and add a few references at this point, including the ones suggested by the referee.

The relevant added section reads: “The experimental realization of extremely short XUV pulses by means of high-order harmonic generation (HHG) using few-cycle VIS/NIR pulses sparked recent experimental and theoretical investigations of such dynamics in cationic molecular systems, for instance in CO_2^+ [13] and small hydrocarbon ions [14-16].”

2) The authors derive the experimental cross-correlation width by fitting the experimental data. They state at page 3 that they used a “biexponential decay function, convoluted with a Gaussian function, representing the experimental time resolution,...”. Later they write that “The experimental cross-correlation width is also found from the fit and amounts to $\tau_{xc} = 10 \pm 1$ fs.” How do they extract the value of pump-probe cross correlation from the Gaussian width? Or in other words, how

do they estimate the impulsive response of the system? Since they show that the probe process is a two-photon process, what the authors get is not strictly a cross-correlation between pump and probe. Can they provide an independent measurement for that?

It is indeed true that the term cross-correlation is not fully accurate in our case since the molecular response is a 1+2' process. Accordingly, we have replaced the word 'cross-correlation' with 'instrument response function' in the manuscript.

The VIS/NIR pulse duration was first optimized to 6fs by means of SEA-F-SPIDER measurements, using a beam path that resembled the beam path to the interaction region. Using glass wedges for dispersion control, the two arms of the setup were then fine-tuned: (i) the VIS/NIR arm was optimized using argon strong field ionization yield (for a less attenuated beam), and (ii) the VUV arm was optimized using HHG efficiency and spectrum (see answer to next question). While explicit data for the GVD of Indium in the VUV frequency range is not available, the effect of a 200nm thick filter on a 3.3fs VUV pulse duration (see next question) is expected to be negligible. On the other hand it is possible that spectral dispersion in Xe plays some role both for the VIS/NIR and the VUV pulse in the HHG arm and/or that the compression of the VIS/NIR pulse was slightly non-optimal.

The multiphoton nature of the molecular response and potential propagation effects of the VIS/NIR and XUV pulses led us to determine the instrument response function from the fit to the experimental data. Note that our data cannot be fitted convincingly by the combination of a cross-correlation function and a single exponential decay. In the manuscript we have clarified the optimization procedure in the methods section and have described why the instrument response function was determined from the fit of the experimental data.

The relevant sections in the manuscript read now:

“Due to the multiphoton nature of the molecular response and potentially not fully compensated pulse propagation effects, we determined a Gaussian instrument response function from a fit of the delay-time dependent ion yield. This resulted in a value of $\tau_{\text{IRF}} = 10 \pm 1$ fs. Note that our data cannot be fitted convincingly by the response function and a single exponential decay.”

and

“Few-cycle pulses with a bandwidth of 640-920nm (FWHM) and a pulse duration of $\tau = 6$ fs, determined with a SEA-F-SPIDER measurement, were obtained. The beam is split into two parts. (...) Using glass wedges for dispersion control, the two arms of the setup were then fine-tuned: (i) the VIS/NIR arm was optimized using argon strong field ionization yield, and (ii) the VUV arm was optimized using HHG efficiency and spectrum.”

3) I understand that the Indium filter cuts the radiation at energies above 17 eV in figure 2.a), but why there is no harmonic ~3 eV below the 9th harmonic in the figure? From the methods one sees that the authors used a Ti:Sa laser which normally is centered around 800 nm. Which is the central wavelength in this case? I ask this because if I assume a harmonic energy of ~ 16 eV for the 9th order I get a central wavelength of ~ 697 nm which is very close to the left border of their IR spectrum (640-920 nm). Is this correct? Could it be that the indium filter cuts the high energy part of the harmonic thus shifting its center of mass down to 16 eV? In this case, is it the 9th or the 11th harmonic that we are looking at? This could also explain the apparently small harmonic bandwidth. With a generating IR pulse of 6 fs one should get an even shorter envelope of the generated attosecond pulse train. I would expect something around 3 fs. So the harmonics in the spectrum should look quite broad. Can the authors comment on this?

The central positions of the HHG peaks in Xe-generated HHG spectra that we recorded without the In filter in place suggest that the utilized harmonic is indeed HH9. We confirmed that the spectral width of this harmonic is unaffected by introducing the filter. From the time-bandwidth product, we calculate a Fourier transform limited pulse duration of 3.3fs, which seems reasonable given the 6fs IR driver.

Since a clean experiment requires monochromatic VUV light, we optimized the HHG phase matching conditions for maximum contrast of the harmonic transmitted through the In filter, at appreciable flux. The procedure resulted in a HHG comb upshifted in energy and allowed us to eliminate lower transmitted harmonics almost entirely. The transmitted harmonic was not significantly spectrally cut by the filter. The central driving wavelength indicated by the referee is well within the bandwidth of the fibre-broadened VIS/NIR pulse.

We are very confident of our frequency calibration, since it is obtained by absorption of the VUV in nitrogen, producing sharp lines as described in the SOM (see Fig.S3).

4) How do the authors calibrate the pump-probe delay? Do they use a different fragmentation channel or the same?

An independent way to bring us close to zero time delay which we used is via exploiting hydrogen bond-softening. However, the Gaussian instrument response function is ultimately determined from the fragment yield as described in the answer to question 2. The fit also yields zero time delay.

The pump-probe delay is generated with a retroreflector actuated by a piezoelectric device. The latter is a rather sophisticated commercial nanopositioning unit, which features an integrated positioning sensor, calibrated shortly before the experiment took place. We rely on the position measurement for calculating the pump-probe delay.

5) From the manuscript one understands that Q_{eff} is a vibrational coordinate. Actually which one? Can the authors give a more intuitive picture of how the nuclei are moving?

The effective coordinate is a suitable linear combination of all normal mode coordinates exhibiting a substantial first-order coupling to the electronic motion (the C-C stretching mode v_2 and the Jahn-Teller active normal modes v_{16} - v_{18}). It is chosen to yield a representative overview over low-energy conical intersections between the relevant electronic states. In the two-state case, simple closed-form expressions can be given that minimize a seam of conical intersections (ACP1984). Then Q_{eff} denotes the direction from the origin to the seam minimum. For more than two states as are relevant here, a coarse-grained procedure is adopted because the intersections involve different pairs of electronic states which cannot be minimized simultaneously. We added this description as a footnote in the figure caption of figure 1 in the manuscript.

(ACP1984): H. Köppel, W. Domcke and L. S. Cederbaum, Adv. Chem. Phys. 57, 59 (1984)

Reviewer #2 (Remarks to the Author):

This article describes new experiments which exploit ultrafast laser pulses in the XUV region to characterize the sub-picosecond dynamics of benzene cation. The primary importance of this work is that it demonstrates that even for very fast (<20fs) processes, one cannot neglect the nuclear motion. This is especially relevant given numerous discussions about ultrafast charge migration

induced by attosecond pulses (which is often modeled theoretically without consideration of nuclear motion). The experiments are certainly state of the art. The agreement with theory is quite impressive, although the authors do make it clear that the interpretation (equating diabatic populations with experimental signals) is not entirely clear. Probably the most annoying part of this article is the paragraph where the authors praise the previous theoretical benzene cation work (last half of the second paragraph of the paper). Since one of the paper's authors is behind that work, this seems immodest at best. The authors should figure out how to discuss that work more objectively. Otherwise, the article is exceptionally well-done and is sure to stimulate further experiments and theory.

We thank the referee for the positive assessment of our work. The paragraph that irritated the referee is a direct result of the mutual admiration that the experimentalists and theoreticians involved in this project developed for each other in the course of the work that was performed. We agree that some of the formulations that were chosen in the original submission were unfortunate. At no point in time was there an intention to brag or be self-serving. We have toned down the second half of the second paragraph of the manuscript considerably to address the referee's point.

Reviewer #3 (Remarks to the Author):

This paper presents an impressive, state-of-the-art pump-probe mass spectrometry study of excited state dynamics in the benzene cation. An approx. 0.5 eV bandwidth 16 eV pump pulse prepares a set of cation states, ranging from X to E (over 5 eV). If transform limited, this pulse would be ~ 4 fs in duration. A ~ 6 fs NIR pulse is used as the probe and the fragmentation spectrum varies with time delay. In particular, the C₄H₃⁺ channel shows pump-probe contrast, as seen in Fig.2(b). The cross correlation is about 10 fs. Through a series of arguments, the authors claim that they probe only the higher E and D electronic states of the benzene cation and that they do so via a two-photon probe: i.e. the C₄H₃⁺ signal arises from a (1+2') process. A probe laser power study is used to support this claim. A state-of-the-art MCTDH calculation is used to model the results and the theoretical electronic population dynamics are compared to the (1+2') pump-probe signal. The agreement looks good, despite that there is no obvious direct link between the calculated population dynamics and the time-resolved C₄H₃⁺ fragment ion yield. Strong claims are made that the fragment ion yield maps the electronic population dynamics. However, this is not proven and could be just hopeful thinking.

There are several issues which need to be clarified before a recommendation for publication can be determined.

1. It took me some time to understand – I hope correctly - that the two time constants of 11 fs and 110 fs are for a parallel decay mechanism, meaning that the E and D states are prepared simultaneously (in a 1:2 ratio, they claim) but decay separately, each with their own time constant. If this is correct, then Fig.1 is highly misleading, as is some of the discussion. Fig. 1 leads the reader to believe that the E state decays to the D state which subsequently decays to the B-state. The reader thinks that the authors are reporting the probing of a sequential decay mechanisms E->D->B.
2. Given that the authors fit to two time constants to a parallel decay, where does the initial E-state population go? I assume to the D state, in which case there should be two components to the D-state decay. One direct preparation via D-state XUV absorption and then subsequent decay and a second path via E-state XUV absorption and then ultrafast decay to the D-state which then undergoes a slower decay. In other words, are there not both parallel and sequential components to the observed D-state decay?

(reply to 1. and 2.):

Indeed the D-state is both directly populated by the XUV and indirectly through decay of the XUV-populated E-state. That's why we construct a superposition of wavepacket calculations initiated in the E- and the D-state for the comparison with the experiment. Under the ergodic assumption the rate equations for the case where you start "parallel" at different points of the single sequential relaxation chain are solved by a double-exponential decay with the same time constants but different prefactors that would result from a pure preparation of the higher lying state. Hence the double-exponential fit model applied to extract timescales from the experimental data is suited.

To clarify this aspect in the manuscript, we have introduced a dedicated third paragraph of the Discussion which reads:

"Note that in our experiment the lower-lying D state is both directly populated by the XUV and indirectly through decay of the XUV-populated E state. The rate equations for relaxation are solved by a double-exponential decay with the same time constants that would result from a pure preparation of the higher-lying E state. We thus can attribute the two exponential time constants observed in the experiment as being associated with relaxation dynamics via the $E \rightarrow D$ and $D \rightarrow B$ conical intersections."

Moreover, in order avoid confusion with the interpretation of Figure 1, we have adapted the caption to the figure, which now reads: "Ionization by the XUV pulse can populate all of the cationic states that are shown (violet arrows). Of these cationic states, only the D-state and the E-state can be excited beyond the threshold for $C_4H_3^+$ formation by a two-photon process (orange arrows)." We have also introduced two small arrowheads in the figure along the decay pathway for further clarification.

3. In Fig.3, the pump-probe $C_4H_3^+$ signal is presented and a biexponential fit is applied. Why is there signal before $t=0$? This is clearly seen in the first 20 fs on the 'negative' time delay side (red bump). Unless the pulse structures are not simple Gaussians as claimed, this means that, somehow, the NIR pulse is absorbed by the parent neutral molecule and, upon absorption of an XUV photon leads to a $C_4H_3^+$ fragment. What is the mechanism by which this occurs? That this channel exists means that the analysis within the 10 fs cross correlation includes both pump-probe and probe-pump processes and that the fit to the 11 fs decay is not so straight forward.

For the neutral benzene molecule, the cross section for absorption of the VIS/NIR pulse is very low. Instead, we attribute the small contribution at negative time-delay to an XUV-VIS/NIR pump-probe signal enabled by an imperfect pulse structure of the 6fs VIS/NIR probe pulse. Laser pulses of such short pulse durations are almost always not perfect gaussians in the time domain, but feature small pre- and post-pulses due to slightly incomplete dispersion compensation. We invest a lot of effort to minimize such effects and use a SEA-F-SPIDER for detailed pulse characterization. Our SEA-F-SPIDER traces from the time of the experiment show a small post-pulse, which is typical. It might account for the noticed small XUV-VIS/NIR signal being generated before the main NIR pulse arrives. None of the findings and conclusions of the manuscript are affected by this artefact.

We have inserted a sentence in the discussion of Fig.3 commenting on the observed artefact: "N.B.: The small contribution observed at negative time delay in Fig.3 probably originates from imperfect dispersion compensation of the VIS/NIR pulse, resulting in a small post-pulse."

4. The relationship between the calculated MCTDH electronic population dynamics and the time dependence of the $(1+2')$ $C_4H_3^+$ signal is not justified. I am worried that the apparent agreement

may be accidental. What is the symmetry allowed two-photon transition from the E and D states? To which 'final' state? Is there only one? Are the two-photon oscillator strengths to the 'final' state the same for the E and D states? Does this not change the way in which the population dynamics should be compared to experiment?

The cation-cation transitions relevant for the probe step are investigated in detail in the Supporting Information. Transition probabilities are computed with the non-Dyson ADC(3) scheme and displayed in Fig.S1. We discuss how fewer final states are reached with significant probability when probing the E-state as compared to the D-state, but that these states are higher in energy and lead with higher probability to fragments. Since this part is rather technical and since the convolution of transition probabilities, photon spectra and fragmentation efficiency depends very sensitively on the quantitative details of each of those we decided that the SOM is the appropriate place for this discussion. The essence is given in the main text and gives support to our interpretation of the experimental findings that the overall detection efficiency is about twice as high for the E-state as for the D-state.

We have made the reference to the SOM more explicit in the main text, and specifically point to figure S1.

5. How does the 'final' state prepared by (1+2') fragment to C₄H₃⁺? How do the Franck-Condon factors between the XUV prepared E and D-states and the 'final' state select the probing of the initial wavepacket? Prior studies by FF Crim et al used time-resolved photodissociation to probe wavepacket dynamics (see for example Journal of Chemical Physics 107, 661 (1997)) and interpreted their results in terms of the evolution of Franck-Condon overlaps. How can the authors avoid such effects? Does not vibrational dynamics in the E and D-states affects the C₄H₃⁺ yield? If so, then the C₄H₃⁺ yield cannot reflect only electronic population dynamics and the agreeable comparison with MCTDH populations is mostly lucky.

The unimolecular dissociation of benzene cations at variable internal energy has been studied in detail by Schlag, Neusser and coworkers [Ref. 28 in the initial submission and Dietz et al., Chem.Phys. 66, 105 (1982)]. Unimolecular rate constants were measured for the primary dissociation channels and found to be in excellent agreement with RRKM rates. Unimolecular dissociation of excited benzene cations is statistically driven, where the molecule relaxes to a vibrationally hot electronic ground state and then falls apart. According to Schlag, Neusser and coworkers, C₄H₃⁺ fragments are produced in a two-step process, where excited C₄H₆⁺ ions first fragment to form C₄H₄⁺, which then loses a hydrogen atom to form C₄H₃⁺. Holland et al. [Ref. 22] also consider another mechanism, which produces C₄H₃⁺ directly from excited C₆H₆⁺. Knowledge of the details of the unimolecular dissociation is of interest, but not essential within our work.

The second point raised by the referee (time-dependence of Franck-Condon overlaps) is indeed a valid concern since this will convolute the time-dependent D/E-state population dynamics that we are interested in with a time-dependent probe laser excitation efficiency. We expect that the cationic wavepackets that are produced by the XUV pump pulses will be spread over several vibrational states. Furthermore, the broad-band probe pulses will then excite the wavepacket to several final vibrational states. Both aspects lead to extensive averaging over the coordinate-dependence of the Franck-Condon factors, leading to a situation where we feel confident neglecting this effect at the current level of detail in the experiment. In other words, we feel that the agreement between experiment and theory is not only "luck", but reflects the fact that the theoretical treatment has captured the essence of the experiment.

We agree with the referee that the coordinate dependence of Franck-Condon factors can play an

important role in certain experiments, in particular if energy is deposited in a specific vibrational mode and the molecule undergoes IVR. Indeed, in the mentioned work by Crim et al., an overtone of the O-H stretch vibration was excited in HNO₃.

Nevertheless, if such an effect as suggested by the referee would contribute, we expect that the slower time scale associated with the D-state decay would be affected more, since the spatial displacement of the wavepacket from the point of creation is larger. Indeed we observe that the time constant from the simulations is smaller than observed in the experiment for the D-state decay. While we believe that the underlying cause is the lack of inclusion of all vibrational modes in the simulation, we have inserted a comment into the main text stating that a time-dependence of the Franck-Condon factors might also contribute.

6. What are the estimated RRMK C₆H₆⁺ → C₄H₃⁺ decay rates at the (1+2') level? Is it obvious that this fragment will appear rapidly enough in the extraction region to appear as a sharp mass peak? This would be yet another 'filtering' of the pump-probe signal. This needs better justification.

We do experimentally observe the C₄H₃⁺ fragment as a well-distinguished delay-dependent mass peak in the time-of-flight spectrum, showing no evidence of the dissociation timescale being comparable to acceleration and flight time (see Fig.2). However, it is true that the total available internal energy is not much higher than the C₄H₃⁺ appearance energy. It is hence indeed possible that the detection efficiency for direct D-state population, where more of the XUV energy is taken away by the photoelectron in the ionization step, is lower than for indirect population of the D-state via the E-state, where the total internal energy is higher. This does however only affect the relative amplitudes in the double-exponential decay, not the time constants. The scenario is captured by our analysis, since we keep the relative detection efficiency of D- and E-state population an adjustable parameter in the comparison with theory.

We have clarified in a footnote in the discussion part that the detection efficiencies for the D-state might depend on the way that the state is formed and that this does not affect the extracted time constants. The footnote reads: "Note also that D-state population, which stems from decay of the E state, might have a higher detection efficiency than directly XUV-populated D-state population. The total internal energy is larger in the first case, which might lead to more efficient formation of the signature fragments. Such an effect would however not affect the timescales extracted in this work."

7. In Fig.4, inset, the MCTDH result (blue line) shows a bump at about 50 fs which is not seen at all in the experimental C₄H₃⁺ yield. Why is that? This should cause some concern that that experiment does not probe just the electronic population dynamics, as claimed.

We have performed computationally very expensive adiabatic calculations in the meantime. These computations agree with the diabatic calculations, but remove some of the inherent artefacts of the diabatic state basis. These features likely emerge from variations of the adiabatic-to-diabatic mixing angle, observed also in previous two-state calculations [Manthe & Köppel]. In our manuscript, we show now the results of both calculations alongside each other (see Fig.4) and describe them in the text. None of the findings and conclusions of the manuscript are affected, but some artefacts of the diabatic state basis including the bump at 50fs disappear. We are thus confident that the electronic populations are at least closely related to the experimental signal (see also item 4 of the reviewer).

[Manthe & Köppel]: Manthe, U. and Köppel, H., Chem. Phys. **93**, 1658 (1990)

Reviewers' comments:

Reviewer #1 (Remarks to the Author):

The present version of the manuscript constitutes a considerable improvement if compared with the previous section. The authors managed to address almost all my points in a convincing manner. Therefore I can now support the publication of the present manuscript in Nature Communication.

Reviewer #3 (Remarks to the Author):

In my view, the revised manuscript has addressed the major concerns with this submitted paper. The technical issues (instrumental time response, pulse spectra) have been addressed, particularly in the revised SOM.

The field of ultrafast measurement science, often dominated by a technical viewpoint, needs to move beyond merely the observation of extremely fast processes. It must be demonstrated that such measurements are both useful and important for advancing broader fields of research, in this case molecular dynamics. Therefore, I do not accept as a criticism that 'faster processes have been observed', nor do I accept that some prior 'stand-alone' theoretical results were published. Furthermore, excited state molecular processes always involve the coupled dynamics of electronic and nuclear motions, even on the 10fs time scale: the field of attosecond science cannot ignore this fact if it addresses timescales longer than a femtosecond (which has always been the case to date). In the present work, it is the powerful combination of state-of-the-art experiment with the same in coupled electronic-nuclear theory (adiabatic MCTDH electronic populations) which has led to a significant advance in molecular dynamics.

Despite their rebuttal, there remain issues to be fixed.

Fig. 1 shows the expected wavepacket dynamics. It still looks to the reader as if there is only a sequential process E->D-> g.s. Since the authors agree that this is misleading, can they please now fix both the figure and the caption.

CAPTION: Please state in the caption:

"Ionization by the XUV pulse can *simultaneously* populate all of the cationic states that are shown"

FIGURE: Please use a green dashed line for the dynamics originating in the E-state (as shown) and *use a different colour* (e.g light blue) for the dynamics originating in the D-state. I.e. you need to use two different dashed lines to depict the dynamics you are studying. Otherwise it looks as if everything starts only from the E-state, or that any dynamics in the D-state follow the same path as those of the E-state.

For the paragraph beginning with "It should be noted that the link between the electronic state populations and the experimental observables is indirect." The authors need to explicitly state their very strong assumptions about the relationship between time-dependent photofragment yield and electronic population dynamics.

Can they now explicitly state that they ignore any vibrational dynamics within the E and D states in terms of its influence on time-dependent photofragment yields. They might suggest that because many internal coordinates are involved, this strong approximation is justified. However, the fairest thing to do would be to admit that since they don't know the internal coordinate dependence of the time-dependent photofragment yield, they have no choice but to assume it is coordinate independent. Only in this limit can they directly compare time-dependent photofragment yields with the MCTDH electronic population dynamics. Can the authors please explicitly state this assumption in the text, ideally at the beginning of this paragraph. An important point such as this should never be hidden from the reader.

If these points are addressed in the final version, I would then consider the manuscript acceptable for publication in Nature Communications.

We would like to thank the referees for their positive assessment of our work and their recommendation to publish our manuscript in Nature Communications.

We have accommodated the two remaining points raised by reviewer #3 by implementing the specific changes that were suggested.

We are positive that our revised manuscript will now be suitable for publication in Nature Communications. Please find our point-by-point response to the reviewer as well as a revised version of our manuscript and a version indicating the changes made enclosed.

Reviewers' comments:

Reviewer #1 (Remarks to the Author):

The present version of the manuscript constitutes a considerable improvement if compared with the previous section. The authors managed to address almost all my points in a convincing manner. Therefore I can now support the publication of the present manuscript in Nature Communication.

Reviewer #3 (Remarks to the Author):

In my view, the revised manuscript has addressed the major concerns with this submitted paper. The technical issues (instrumental time response, pulse spectra) have been addressed, particularly in the revised SOM.

The field of ultrafast measurement science, often dominated by a technical viewpoint, needs to move beyond merely the observation of extremely fast processes. It must be demonstrated that such measurements are both useful and important for advancing broader fields of research, in this case molecular dynamics. Therefore, I do not accept as a criticism that 'faster processes have been observed', nor do I accept that some prior 'stand-alone' theoretical results were published. Furthermore, excited state molecular processes always involve the coupled dynamics of electronic and nuclear motions, even on the 10fs time scale: the field of attosecond science cannot ignore this fact if it addresses timescales longer than a femtosecond (which has always been the case to date). In the present work, it is the powerful combination of state-of-the-art experiment with the same in coupled electronic-nuclear theory (adiabatic MCTDH electronic populations) which has led to a significant advance in molecular dynamics.

Despite their rebuttal, there remain issues to be fixed.

Fig. 1 shows the expected wavepacket dynamics. It still looks to the reader as if there is only a sequential process E->D-> g.s. Since the authors agree that this is misleading, can they please now fix both the figure and the caption.

CAPTION: Please state in the caption:

"Ionization by the XUV pulse can *simultaneously* populate all of the cationic states that are shown"

FIGURE: Please use a green dashed line for the dynamics originating in the E-state (as shown) and *use a different colour* (e.g light blue) for the dynamics originating in the D-state. I.e. you need to use two different dashed lines to depict the dynamics you are studying. Otherwise it looks as if everything starts only from the E-state, or that any dynamics in the D-state follow the same path as those of the E-state.

We inserted an additional light blue arrow into Fig.1 as requested by the reviewer and updated the caption accordingly to clarify that there are two pathways.

For the paragraph beginning with “It should be noted that the link between the electronic state populations and the experimental observables is indirect.” The authors need to explicitly state their very strong assumptions about the relationship between time-dependent photofragment yield and electronic population dynamics.

Can they now explicitly state that they ignore any vibrational dynamics within the E and D states in terms of its influence on time-dependent photofragment yields. They might suggest that because many internal coordinates are involved, this strong approximation is justified. However, the fairest thing to do would be to admit that since they don't know the internal coordinate dependence of the time-dependent photofragment yield, they have no choice but to assume it is coordinate independent. Only in this limit can they directly compare time-dependent photofragment yields with the MCTDH electronic population dynamics. Can the authors please explicitly state this assumption in the text, ideally at the beginning of this paragraph. An important point such as this should never be hidden from the reader.

We inserted two sentences into the indicated paragraph in the Discussion section stating and justifying our approximation. The two sentences read: “In our analysis, we ignore any potential influence of vibrational dynamics within the E and D states on the probing efficiencies. We justify this approximation by the extensive averaging over the coordinate-dependence of Franck-Condon factors due to the many internal coordinates involved.”

If these points are addressed in the final version, I would then consider the manuscript acceptable for publication in Nature Communications.

REVIEWERS' COMMENTS:

Reviewer #3 (Remarks to the Author):

In this revised version, the authors have addressed all my concerns about this manuscript. In my view, the paper should now be acceptable for publication in Nature Communications.